# Studies of Selective Recovery of Zinc and Manganese from Alkaline Batteries Scrap by Leaching and Precipitation

**DOI:** 10.3390/ma15113966

**Published:** 2022-06-02

**Authors:** Tomasz Skrzekut, Andrzej Piotrowicz, Piotr Noga, Maciej Wędrychowicz, Adam W. Bydałek

**Affiliations:** 1Faculty of Non-Ferrous Metals, AGH University of Science and Technology, 30-059 Krakow, Poland; skrzekut@agh.edu.pl (T.S.); tankist@onet.eu (A.P.); pionoga@agh.edu.pl (P.N.); 2Institute of Materials and Biomedical Engineering, Faculty of Mechanical Engineering, University of Zielona Gora, 65-516 Zielona Gora, Poland; abydalek@uz.zgora.pl

**Keywords:** alkaline batteries recycling, zinc recovery, manganese recovery, leaching, precipitation

## Abstract

Recovery of zinc and manganese from scrapped alkaline batteries were carried out in the following way: leaching in H_2_SO_4_ and selective precipitation of zinc and manganese by alkalization/neutralization. As a result of non-selective leaching, 95.6–99.7% Zn was leached and 83.7–99.3% Mn was leached. A critical technological parameter is the liquid/solid treatment (l/s) ratio, which should be at least 20 mL∙g^−1^. Selective leaching, which allows the leaching of zinc only, takes place with a leaching yield of 84.8–98.5% Zn, with minimal manganese co-leaching, 0.7–12.3%. The optimal H_2_SO_4_ concentration is 0.25 mol∙L^−1^. Precipitation of zinc and manganese from the solution after non-selective leaching, with the use of NaOH at pH = 13, and then with H_2_SO_4_ to pH = 9, turned out to be ineffective: the manganese concentrate contained 19.9 wt.% Zn and zinc concentrate, and 21.46 wt.% Mn. Better selectivity results were obtained if zinc was precipitated from the solution after selective leaching: at pH = 6.5, 90% of Zn precipitated, and only 2% manganese. Moreover, the obtained concentrate contained over 90% of ZnO. The precipitation of zinc with sodium phosphate and sodium carbonate is non-selective, despite its relatively high efficiency: up to 93.70% of Zn and 4.48–93.18% of Mn and up to 95.22% of Zn and 19.55–99.71% Mn, respectively for Na_3_PO_4_ and Na_2_CO_3_. Recovered zinc and manganese compounds could have commercial values with suitable refining processes.

## 1. Introduction

Zinc-based batteries, such as zinc carbon, alkaline, etc., still have a significant share in the global energy storage market, despite the growing popularity of lithium-ion batteries [1,2]. Regardless of the type of zinc-based batteries, i.e., whether they are non- or rechargeable batteries, as a result of their use for consumer purposes, these batteries are electronic scrap and must be disposed of [3,4]. Due to the chemical composition of these raw batteries, i.e., the high content of zinc and manganese, 8–18% and 26–43% by weight [5], respectively, recycling of these metals can be considered. Separated and recovered battery components may constitute a secondary source of metals [3,4]. In addition to zinc and manganese recycling the recycling of graphite from zinc batteries, among others, is also considered.

Raw materials are crucial to the economy, especially economies of the European Union, which are focused on hi-technology and mega-trends. Raw materials form a strong industrial base, producing a broad range of goods and applications used in everyday life and modern technologies. Reliable and unhindered access to certain raw materials is a growing concern within the EU and across the globe. Technological progress and quality of life rely on access to a growing number of raw materials. For example, a smartphone might contain up to 50 different kinds of metals, all of which contribute to its small size, light weight and functionality. Raw materials are closely linked to clean technologies. They are irreplaceable in solar panels, wind turbines, electric vehicles, and energy-efficient lighting.

Among the methods of Zn-battery recycling, hydrometallurgical methods are the leading ones, due to their many technological advantages, such as the possibility of obtaining pure recovered components of Zn-battery scrap. These methods include black mass leaching and selective zinc and manganese precipitation. There are also more “sophisticated and refined” methods, such as hydrochloric leaching [6,7,8], leaching assisted by sulfur dioxide [9], sulfides precipitation [10], manganese and zinc crystallization [11], separation and purification process by solvent extraction [12,13,14,15], ultrasound- and microwave-assisted leaching [16], electrolysis from post-leaching solution [17,18], processing by acetic-hypochlorite [19], bioleaching [20], etc. However, from a technological and economic point of view, a method based on the leaching-precipitation stages seems to be the most appropriate from the point of view of large industrial processing and economic profit [21,22,23,24,25].

Leaching of black mass in sulfuric acid, followed by selective precipitation of zinc and manganese from the solution is a well-known and practical method of recycling Zn batteries [12,25,26,27,28,29,30,31]. However, as it turns out, the selection of technological parameters of these steps implies successive steps, and, ultimately, the end products may vary depending on the parameters of the steps.

Study on the separation of manganese and zinc from the concentrate of alkaline batteries scrap (the so-called black mass) is the subject of this article. In this study, we used the hydrometallurgical method to obtain pure and commercially valuable compounds. Two different methods of hydrometallurgical treatment were investigated: selective and non-selective leaching. Scheduled treatment by selective leaching, the scheme of which is shown in Figure 1, took place as follows:(1)non-reductive leaching, the purpose of which was to leach out zinc with minimal leaching of manganese;(2)precipitation of zinc by alkalizing a solution mainly containing zinc;(3)reductive leaching of the solid residue in sulfuric acid and the zinc precipitation in Mn solution.

**Figure 1 materials-15-03966-f001:**
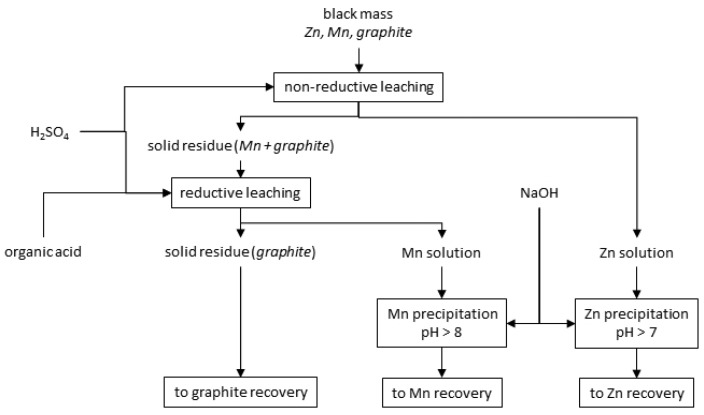
Proposal of hydrometallurgical treatment by selective leaching of alkaline batteries’ black mass for zinc, manganese and graphite recovery.

Regarding the second method, treatment by non-selective leaching (Figure 2), we followed the process below:(1)reductive leaching, the purpose of which was to leach both manganese and zinc and separate them from solid residue (graphite);(2)precipitation of manganese by alkalization of manganese-zinc solution;(3)precipitation of zinc by neutralizing the solution after alkalization.

**Figure 2 materials-15-03966-f002:**
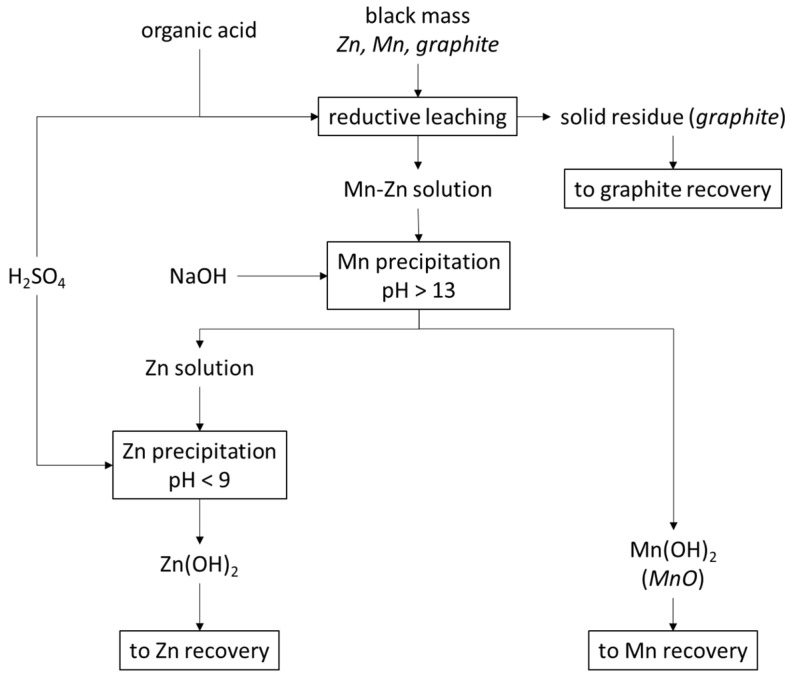
Proposal of hydrometallurgical treatment by non-selective leaching of alkaline batteries’ black mass for zinc, manganese and graphite recovery.

Both methods of processing allow the separation of all valuable components of the black mass: manganese, zinc and graphite.

During the leaching of the black mass with sulfuric acid (non-concentrated), the leaching of mainly zinc occurs, according to the reaction:(1)$$H_{2}{\mathrm{SO}}_{4}{}_{\mathrm{sol}}\to {\mathrm{ZnSO}}_{4}{}_{\mathrm{sol}}+H_{2}O_{\mathrm{sol}}$$

There is also a limited leaching of manganese:(2)$${\mathrm{Mn}}_{2}O_{3}+H_{2}{\mathrm{SO}}_{4}{}_{\mathrm{sol}}\to {\mathrm{MnSO}}_{4}{}_{\mathrm{sol}}+{\mathrm{MnO}}_{2}+H_{2}O_{\mathrm{sol}}$$
(3)$${\mathrm{Mn}}_{3}O_{4}+H_{2}{\mathrm{SO}}_{4}{}_{\mathrm{sol}}\to 2{\mathrm{MnSO}}_{4}{}_{\mathrm{sol}}+{\mathrm{MnO}}_{2}+2H_{2}O_{\mathrm{sol}}$$

Manganese is efficiently dissolved during the reductive leaching that occurs when a reducing acid is used [27,29]—for example, ascorbic acid:(4)$$10{\mathrm{MnO}}_{2}+10H_{2}{\mathrm{SO}}_{4}{}_{\mathrm{sol}}+C_{6}H_{8}O_{6}\to 10{\mathrm{MnSO}}_{4}{}_{\mathrm{sol}}+14H_{2}O_{\mathrm{sol}}+↑6{\mathrm{CO}}_{2}$$

Therefore, depending on whether the leaching occurs with, or without, a reducing agent, there is either a selective (non-reductive) or collective (reductive) leaching, and this stage determines the further steps in the recovery of zinc and manganese. There are various reducing agents with good results for the acid-reductive dissolution of manganese, such as SO_2_, hydroxylammonium chloride [32]. Authors of this study used ascorbic acid, as did other researchers [27,29]. Ascorbic acid has many practical advantages, including being non-toxic, easy-to-use, highly soluble, and having harmless by-products when decomposing.

The theoretical basis of zinc and manganese separation are presented in the diagrams below (Figure 3 and Figure 4). Manganese can precipitate from acidic solution either by changing the pH (alkalization) or changing the oxidative conditions. Under the influence of a strong oxidant agent, e.g., hydrogen peroxide, manganese may precipitate as MnO_2_, regardless of pH_final_ value. Changing the pH from 0 to 4 and above causes the manganese to precipitate in the form of MnO_2_, MnO∙OH and/or Mn(OH)_2_: the higher the pH, the more manganese should be precipitated. Practically, all manganese should be precipitated after exceeding pH of approx. 8. On the other hand, zinc is precipitated after exceeding the pH value of approx. 6. Therefore, on this basis, one could consider the scenario of selective zinc precipitation, which would take place in the range of pH 6–8:(5)$${\mathrm{Mn}}^{2+}+2{\mathrm{OH}}^{-}\overset{\mathrm{pH}=6–8}{↔}{\mathrm{Mn}}^{2+}+2{\mathrm{OH}}^{-}$$
(6)$${\mathrm{Zn}}^{2+}+{\mathrm{OH}}^{-}\overset{\mathrm{pH}=6–8}{\to }\mathrm{ZnO}+H^{+}$$

The separation of zinc from manganese takes place as follows: first, the solution must be neutralized to a final pH = 6–8, as a result of which zinc precipitates; then, manganese precipitates from the solution at a pH above 8. This method is dedicated when there is little manganese in the solution, because in the case of a high concentration of manganese, manganese will co-precipitate with zinc. Therefore, this method of separating zinc from manganese should work for a selective leach treatment. However, this is a “partial” selective precipitation because, as previously indicated, manganese can start to precipitate even at a pH above 4. The less manganese is leached earlier, the less manganese co-precipitation there will be. Moreover, the pH range at which zinc can be separated is relatively narrow (2 units of pH).

At a very high pH value (practically 13–14 [29]), zinc is complexed (not shown in Figure 4):(7)$${\mathrm{Zn}}^{2+}+{\mathrm{OH}}^{-}\overset{\mathrm{pH}=13–14}{\to }{\mathrm{Zn}}_{x}{\left(\mathrm{OH}\right)}_{y}{}^{z-}$$

Since manganese is completely precipitated in the form of Mn(OH)_2_ under these pH conditions, it is possible to separate the manganese from the zinc in this way. In this case, manganese precipitates first, followed by zinc. Thereafter, zinc precipitates from the manganese-free solution, acidifying the solution to a pH value of 6–8 (manganese is in the ionic phase, and zinc is precipitating as ZnO). Thus, the separation of zinc from manganese in this way is as follows: first, the solution is made alkaline to a high pH, at least above pH 8, the precipitated manganese is separated from the solution, and then zinc is precipitated from the manganese-free solution by acidifying the solution to pH 6–8. Due to the need to achieve a high pH, a high consumption of alkalizing agent, for example, sodium hydroxide, is expected. For zinc precipitation, sulfuric acid can be used to neutralize the alkaline solution. As a result of the high concentration of sodium hydroxide solution and sulfuric acid, sodium sulphate may precipitate:(8)$$2{\mathrm{Na}}^{+}+{\mathrm{SO}}_{4}{}^{2-}\to {\mathrm{Na}}_{2}{\mathrm{SO}}_{4}$$

While Mn(OH)_2_ decomposes under the influence of air, zinc concentrate requires either a high-temperature treatment, due the presence of co-precipitating Zn(OH)_2_, or an acidic hydrometallurgical treatment, i.e., solution in e.g., sulfuric acid.

As a result of the proposed alkaline batteries’ black mass processing, the following products are obtained: MnO_2_, ZnO, graphite and Na_2_SO_4_ as by-product. They all have commercial value or are suitable for enrichment processes. Manganese (IV) oxide is a widely used reagent (strong oxidizer, component of batteries and capacitors, etc. [34,35,36]) that can be also further processed to obtain metallic manganese. Zinc oxide is an intermediate in zinc metallurgy [37,38]. Zinc oxide is easily leached in sulfuric (VI) acid, from which electrolytic zinc can be obtained. Graphite is a critical raw material for European Union economies [39,40], hence the need for its recovery; according to [38], graphite is currently not considered a by-product of Zn-battery recycling, although it might be. Sodium sulfate is a byproduct of the proposed technological solution.

The aim of the laboratory research was to investigate the recovery of zinc and manganese from alkaline batteries scrap by means of a hydrometallurgical method. Part of the research, that dealt with non-selective leaching, is similar in assumptions to those in another study [29]. The effect of the type of leaching, i.e., selective or non-selective leaching in the first stage, on the leaching yields of manganese, zinc and iron from the black mass of alkaline batteries scrap was investigated. In terms of efficiency and selectivity, one of these methods was selected for further research. What is completely scientific novum is the study of precipitation using previously unused reagents, i.e., sodium phosphate and sodium carbonate. Subsequently, the possibility of selective separation of manganese from zinc by changing the pH of the solution and the kind of precipitating agent, which influenced the precipitating yields of zinc, manganese, iron and other elements, were investigated.

## 2. Materials and Methods

The initial material for the study was manganese-zinc-graphite concentrate from cryogenic, magnetic and mechanical processing and separation methods of alkaline batteries’ scrap, i.e., black mass. The following reagents were used for calibration and concentration determinations of ions in solutions before and after leaching or precipitation, and also for the precipitation process: sulfuric acid, sodium hydroxide, manganese (II) sulfate, zinc sulfate, iron (II) sulfate, ascorbic acid, sodium carbonate, sodium phosphate (all from Avantor, Poland). The concentrations of elements in the solid and liquid phases were determined by energy dispersive X-ray fluorescence analysis (ED-XRF, MiniPal4 PANalytical). Microscopic examination of manganese and zinc precipitates, using scanning electron microscope with energy dispersion spectroscopy system (SEM-EDS, SU-70, Hitachi, Tokyo, Japan) was also performed. Based on the results, both the quantitative chemical composition of the solid phase and the leaching or precipitation yields for the individual processing steps were determined. The composition of the initial material was determined from three parallel measurements. The qualitative composition of the solid phases was investigated using X-ray powder diffraction analysis (XRD, Rigaku MiniFlex II). The pH of the solutions was recorded with a pH-electrode type ERH-13-6 (Hydromet, Głogów Małopolska, Poland) by using a CX-741 ELMETRON multimeter. Prior to each pH measurement, the electrode was calibrated using Hamilton Duracal buffers. The mixing was carried out on a magnetic stirrer (LLG-uniSTIRRER7); the rotational speed in all cases was 200 rpm.

### 2.1. Determination of the Amount of Graphite (Insoluble Solid Residue) in the Black Mass

Since the ED-XRF method does not allow the amount of carbon in the black mass to be determined, to determine the amount of graphite (or insoluble solid residue), about 10 g of the initial material was taken up in 200 mL of aqua regia for 2 h. The residue was filtered, weighed and its composition determined. Weight loss due to leaching was defined as graphite and was calculated:(9)$$C_{\mathrm{graphite}}=\frac{m_{0}-m_{1}}{m_{0}}\times 100\%$$
where: C_graphite_ is amount of graphite in black mass, % by mass; m_0_ is mass of the initial material (before leaching in aqua regia), g; and m_1_ is the final mass (after leaching in aqua regia), g.

### 2.2. Selective and Non-Selective Leaching of Black Mass

The leaching yields of manganese, zinc and iron as a result of non-reducing (selective) leaching and reducing (collective) leaching were determined. The course of selective leaching was as follows: 5 g of the initial material was leached with sulfuric acid of various concentrations: 0–0.5 mol∙L^−1^.The following were permanent technological parameters: room temperature, duration—120 min, liquid-solid ratio (l/s)—20 mL∙g^−1^. In parallel to selective leaching, a non-selective leaching study was carried out. The technological parameters were as follows: room temperature, 0.5 mol∙L^−1^ H_2_SO_4_, duration—120 min, ascorbic acid concentration—10 g∙L^−1^. The variable parameter was the liquid/solid ratio. The leaching yields of manganese, zinc and iron were determined, as well as the chemical composition of solid phases after leaching (solid residues).

The mass change of the solid phase as a result of leaching was calculated by:(10)$$∆m_{\mathrm{leach}}=\frac{m_{0}-m_{\mathrm{leach}}}{m_{0}}\times 100\%$$
where: ∆m_leach_ is mass change of the solid phase, %; m_0_ is mass of the initial material (before leaching), g; and m_leach_ is mass of solid residue (after leaching), g. The leaching yields were calculated from the following formula:(11)$$\eta_{X}=\frac{C_{X,1}\times V_{1}}{C_{X,0}\times m_{0}}\times 100\%$$
where: η_X_—leaching yield of X, %; C_X,1_—concentration of X in solution after leaching, g∙mL^−1^; V_1_—volume of solution after leaching, mL; C_X,0_—concentration of X in initial material, % by mass; m_0_—mass of the initial material, g; X—element: Mn, Zn or Fe.

### 2.3. Manganese and Zinc Precipitation

First, prior to each precipitation test, the initial pH of the solution after leaching was determined. Then, 100 mL of the solution was made alkaline to the adjusted pH value. The amount of the precipitating reagent: NaOH, H_2_SO_4_, Na_2_CO_3_ and Na_3_PO_4_ was determined to achieve the adjusted pH. The precipitates were calcined at 500 °C in an electric furnace: 7.5 h—time to reach the calcination temperature; 1.5 h—calcination duration. The calcined precipitates were subjected to quantitative and qualitative analysis. On this basis, it was possible to determine the optimal pH value for the manganese and zinc precipitation process.

The precipitation yields were calculated from the following formula:(12)$$\mu_{X}=\frac{\left(C_{X,1}\times V_{1})-(C_{X,2}\times V_{2}\right)}{C_{X,1}\times V_{1}}\times 100\%$$
where: μ_X_—precipitation yield of X, %; C_X,1_—concentration of X in solution after leaching, g∙mL^−1^; V_1_—volume of solution after leaching equals 100 mL; C_X,2_—concentration of X in solution after precipitation, g∙mL^−1^; V_2_—volume of solution after precipitation, mL; X—element.

## 3. Results and Discussion

### 3.1. Chemical Composition of Black Mass

Table 1 shows the percentage of insoluble solid residue in the initial material. It constitutes about 6% of the black mass. Table 2 shows the chemical composition of the initial material and the insoluble solid residue. The black mass consists mainly of zinc and manganese (approx. 41 and 43 wt.%, respectively, in total approx. 90 wt.%). Another significant component of the black mass is potassium, approx. 6.5 wt.%, which is part of the electrolyte in alkaline batteries. The insoluble solid residue consists of a small amount of zinc and manganese and about 72% chlorine. This chlorine content is related to the presence of the insoluble ammonium chloride contained in batteries scrap. Due to the fact that it is not possible to determine the amount of carbon (graphite) in the insoluble solid residue by the ED-XRF method, it was assumed that it makes up most of it and will be referred to simply as graphite [see Table 2—composition of the black mass, including the insoluble solid residue (graphite)].

### 3.2. Selective and Non-Selective Leaching of Black Mass

The leaching of manganese and zinc from black mass was tested in two ways: without and with the addition of ascorbic acid (reductive agent for manganese). The purpose of the leaching test without the addition of a reducing agent was to examine whether it is possible to selectively leach zinc without leaching manganese.

Figure 5 shows the relationship between the mass change of the solid phase (∆m_leach_) and the leaching yields (η_X_) of zinc and manganese, depending on the concentration of sulfuric acid. As a result of leaching, 10–60% of the solid phase was leached, depending on the concentration of sulfuric acid: 0–0.5 mol∙L^−1^. Leaching with dilute sulfuric acid (0–0.25 mol∙L^−1^) causes the majority of zinc, up to approx. 90%, to dissolve while maintaining a minimum manganese dissolution, not more than 1%. Increasing the concentration of sulfuric acid partially dissolves the manganese, therefore selectivity is no longer maintained. Thanks to the appropriate conditions, no iron leaching was registered in any case (concentration of iron in the solution was below the level of determination). Between ∆m_leach_ and η_Zn_ in the range of C_H2SO4_ = 0–0.5 mol∙L^−1^, there is a close linear relationship, R^2^ = 0.9838 (Figure 6). Hence, manganese leaching only contributed to a minor extent to ∆m_leach_.

Table 3 shows the chemical composition of the solid residues after leaching without reducing agent. As the concentration of sulfuric acid increases, the concentration of manganese and iron increases, to approx. 91 and approx. 3 wt.%, respectively, and the concentration of zinc in the solid residues decreases, to approx. 1.5 wt.%, which confirms the effectiveness of the selective leaching. The material consists mainly of manganese oxides of various oxidation states (Figure 7). MnO_2_ is slightly soluble in sulfuric acid, and Mn_2_O_3_ does not dissolve. In water (0 mol∙L^−1^), potassium, chlorine and ammonium salts can be dissolved from black mass. The solution after leaching with water was strongly red, confirming the above claim. With regards to potassium, a decrease in the content of this element in the solid phase was noted, from approx. 6.5 to 1% by mass (compare Table 2 and Table 3).

The above results show that manganese can be separated from zinc by zinc leaching from black mass. It is a simple method, requiring only low concentration of sulfuric acid (up to 0.25 M). The mixture of manganese oxides in the solid residue requires further treatment, for example, by reductive leaching. These results provide the basis for further research in this scientific area.

As an alternative to selective leaching, non-selective leaching tests were carried out with the use of the manganese reducing agent, i.e., ascorbic acid. During the leaching, gas bubbles intensely evolved, which indicated that the ascorbic acid was oxidizing (CO_2_ evolution). The ratio of liquid to solid phase (l/s) was adopted as the critical technological parameter, which reflected the scale and profitability of recycling alkaline batteries by the hydrometallurgical method. Figure 8 shows the effect of the solid phase mass loss and the zinc and manganese yields, depending on the liquid-solid ratio. The following conclusions can be drawn from Figure 8: mass loss is very high, to almost 100%; the optimal liquid-solid ratio value is 20 mL∙g^−1^, above this value of liquid–solid ratio, the manganese and zinc leaching yields are equally high; it is possible to leach manganese and zinc to almost 100% by reductive leaching; iron does not dissolve. The high value of liquid–solid ratio, equaling 20 mL∙g^−1^, raises doubts about the use of this method in industrial practice. Another inconvenience is the emission of CO_2_ (see Equation (4)), a greenhouse gas. The European Union is trying to reduce emissions of greenhouse gases [41]. The advantage of this method of leaching is bringing the manganese and zinc into the liquid phase, which facilitates their further processing.

Table 4 shows the composition of the solid residues after reductive leaching for different liquid–solid ratio ratios. As the liquid–solid ratio increases, the concentrations of manganese and zinc in solids decrease, to approx. 4 and 1.5 wt.%, respectively, and the concentrations of iron, barium and sulfur increase; the exception is liquid–solid ratio = 40 g∙L^−1^, for which the zinc and manganese contents increase to approx. 12 and 11% by mass, respectively, compared to liquid–solid ratio = 30 g∙L^−1^. Iron does not dissolve and its content in solid residue is from about 1 to about 33% by mass; the sulfur content is likely to be due to the precipitated sulphate salts; the presence of barium can be explained as the presence of scrap electrical elements containing perovskites (barium compounds) [42]—the barium content reaches up to about 40% by mass. The manganese contained in the residue is primarily insoluble Mn_3_O_4_ (Figure 9), which cannot be leached under these technological conditions. Please note that the residue also includes graphite which, due to the composition determination method, cannot be determined.

Reductive leaching allows manganese and zinc to be leached in one technological step. A slight addition of ascorbic acid, in the amount of 10 g per liter, compared to leaching without this acid, significantly changes the leaching conditions, and, therefore, also the leaching values, especially for manganese. Practically all zinc and manganese are dissolved. This implies that in the next step the manganese and the zinc have to be separated from the solution.

Summarizing, selective leaching, in which the zinc is leached and the manganese not, should be carried out with a sulfuric acid concentration of 0.25 mol∙L^−1^; increasing this value to over 0.25 mol∙L^−1^ causes manganese collective leaching. When it comes to non-selective leaching, it takes place under similar conditions to selective leaching, but with the addition of 10 g∙L^−1^ ascorbic acid. An important leaching parameter is the phase ratio—it should be at least 20 mL∙g^−1^. Under optimal parameters, the leaching yields for zinc and manganese are approx. 85 and approx. 1%, and 99.2% and 83.7%, for selective and non-selective leaching, respectively.

### 3.3. Zinc and Manganese Precipitation from Selective or Collective Treatment Solutions

The next stage of the study was the precipitation of zinc and manganese from solutions after selective and non-selective leaching. Both solutions require alkalization in order to separate both elements. For this purpose, the effects of pH on the precipitation yields of zinc and manganese were investigated. The necessary amounts of the precipitating agent to achieve the adjusted pH value and the chemical composition and microscopic pictures of the obtained precipitates were determined.

The effect of sodium hydroxide as a precipitating agent from the solution after selective (Figure 1) leaching is presented in Table 5, which shows the amounts of NaOH needed to make the solution alkaline to the adjusted pH (m_NaOH_/V_sol_), and includes the amount of solid phase precipitated (∆m_prec_/V_sol_)) Yhe graphs below are presented on its basis: Figure 10 and Figure 11. Figure 12 shows the precipitation yields of zinc and manganese. Up to pH = 5.5 and 6, there was little precipitate, i.e., 1.0–1.6 g∙L^−1^, but at pH = 6.5, the amount of the precipitate was 13.6 g∙L^−1^ (Figure 10). Further alkalization did not, practically, contribute to any further gain in amount. The substantial amount of NaOH needed to reach pH = 6.5 indicates a precipitation process (Figure 11). Figure 12 shows that at pH = 6.5, more than 90% of zinc and a little manganese (approx. 2%) precipitated. The increase in pH led to a slight increase in zinc precipitation yield, i.e., to a value of approx. 95%, and to an increase in manganese precipitation yield, i.e., 8–47%. Thus, the best pH value to strive for is 6.5 as it allows most of the zinc to precipitate, while maintaining a low manganese precipitation yield.

The precipitates were yellowish white when obtained at pH = 5.5–6, and white at pH = 6.5; blackening of precipitates due to air was observed with increasing pH. Figure 13 shows the chemical composition of the precipitates and Figure 14 the relationship between adjusted pH and the amounts of precipitated elements (m_X_/V_sol_, g∙L^−1^). The precipitates obtained at pH = 5.5–6.5 had a high concentration of zinc and a low concentration of manganese, i.e., 90–91 and 1.0–1.1 wt.%, respectively. An increase in pH caused an increase in manganese content at the expense of zinc, 4–18 and 88–76 wt.%, respectively. In the range of pH = 5.5–10, the concentration of SO_3_, resulting from the co-precipitation of sodium sulphate, was constant and amounted to 5.5–6.5 wt.%; at pH = 11.5 it amounted to approx. 8 wt.%.

Figure 15 shows the precipitates obtained at pH = 6.5 with a high concentration of zinc and a low concentration of manganese, i.e., 91 and 2 wt.%, respectively. The enlarged area (Figure 15) indicates the propagation of zinc concentrate in the areas of NaOH lumps. The remaining gray-black-black areas are fragments of Zn-Mn solution with NaOH mixture—a kind of eutectic mixture.

Precipitation studies were performed using NaOH as the precipitating agent also for the solution after non-selective leaching (Figure 2). Table 6 shows the amounts of NaOH needed to make the solution alkaline to the adjusted pH and includes the amount of solid phase precipitated, and on its basis the following graphs are presented: Figure 16 and Figure 17. While the alkalization from the pH of the initial solution, i.e., about 1.4–1.5, to pH = 12 took place with a proportionally increasing amount of NaOH (8–38 g∙L^−1^), the increase in pH from 12 to 13 required much more NaOH, i.e., approx. 130 g∙L^−1^. Up to pH = 10 a precipitate starts to form from the solution, but above 10 the precipitate begins to re-dissolved. The dissolution of the precipitate is attributed to the digestion of the zinc hydroxide towards its complexation. The highest rational pH that was achieved was pH = 13. From Figure 16 it can be concluded that the pH value to be pursued for the precipitation of zinc is 10 or less. Hence, after the manganese is precipitated at pH = 13, the solution should be neutralized to pH = 10.

The freshly precipitated solids turned violet-brown in color and turned black on exposure to air, which indicated that they contained manganese hydroxide. As the pH increased, the concentration of manganese increased (from 8.5 to 66 wt.%) and the concentration of zinc decreased (from 82 to 20 wt.%) (see also Figure 18 and Figure 19). High sulfur concentration in the range of pH = 10–12 was caused by the co-precipitation of sodium sulphate, which resulted from the increasing sulfur content, reaching approx. 27 wt.% SO_3_. The precipitate obtained at pH = 13 contained considerable amounts of both manganese and zinc, approx. 66 and 20 wt.%. Such material requires further treatment, e.g., by selectively leaching zinc without leaching the manganese. It is not quite a selective method; moreover, it requires the use of 130 g of NaOH per liter of solution.

There are a number of aspects to consider when comparing the precipitation of zinc and manganese from solution after selective and non-selective leaching with NaOH. Alkalization of the solution after selective leaching to the optimal pH value at which there is a high precipitation yield of zinc in relation to manganese, i.e., pH = 6.5, required much less NaOH, only about 10 g∙L^−1^, than the alkalization of the solution after non-selective leaching to the required high pH = 13, i.e., almost 135 g∙L^−1^ NaOH (compare Figure 10 and Figure 16). The chemical composition of the precipitate obtained from the solution after selective leaching at pH = 6.5 was satisfactory, due to the very high zinc content and low manganese content (Figure 13); although the case was similar for the precipitate from non-selective leaching (Figure 18). However, comparing Figure 14 and Figure 19, i.e., regarding the dependencies of the amounts of precipitated elements per unit volume of the solution, considerably more zinc was precipitated from the solution after selective leaching (i.e., about 12.5 g∙L^−1^) than from the solution after non-selective leaching (only about 1 g∙L^−1^). Although the precipitation from the non-selective leaching solution obtained at pH = 13 contained a high manganese content and a relatively low zinc content, of about 65.5 and about 20 wt.%, respectively, it is difficult to speak of a successful selective precipitation in this case. The method of selective zinc leaching and its subsequent precipitation turned out to be a better solution than the method of non-selective leaching and selective precipitation of manganese and zinc, mainly due to the savings on NaOH and obtaining a satisfactorily efficient separation between zinc and manganese in the first-mentioned method. The precipitated zinc from selective treatment is suitable for further processing, such as leaching in sulfuric acid, followed by electrolysis of zinc from zinc sulphate.

A test for selective zinc precipitation by neutralization from the collective treatment (Figure 2) process was also performed. After precipitating the manganese at pH = 13 by using NaOH, zinc was precipitated from the post-processing solution by using sulfuric acid to an adjusted pH = 9. As a result, the solution began to discolor to an intense violet and a white precipitate began to precipitate. Its chemical composition was as follows: 21.46 MnO, 76.66 ZnO, 2.29 SO_3_, 0.02 K_2_O, 0.0911 Fe_2_O_3_ + NiO (wt.%, Figure 20). Due to the high content of manganese, similar to the manganese concentrate obtained in the earlier stage, the zinc concentrate required further stages of zinc concentration. This proved that the recovery of zinc and manganese by the collective method is not very effective (Figure 2), which is in opposition to what has been presented in [29]. This is probably due to the initial material: grain size, chemical composition, etc., the further processing of which influenced all the results of the researches.

Considering the above results of the precipitation of zinc and manganese with sodium hydroxide, further research focused on the precipitation of zinc from the solution resulting from selective leaching.

#### 3.3.1. Sodium Phosphate Buffer

The possibility of using sodium phosphate (Na_3_PO_4_) for alkalization and zinc precipitation was investigated. The laboratory research with the use of sodium phosphate was carried out in analogy to the tests with the use of sodium hydroxide. Table 7 shows the amounts of Na_3_PO_4_ needed to make the solution alkaline to the adjusted pH (m_Na3Po4_/V_sol_) and includes the amount of solid phase precipitated (∆m_prec_/V_sol_). The graphs below are presented on its basis: Figure 21 and Figure 22. Figure 23 shows the precipitation yields of zinc and manganese. Precipitation was observed from about pH 4 and above. At pH > 4.5, there was a little precipitate, approx. 9 g∙L^−1^, but at pH = 4.5, the amount of the precipitated phase was 30 g∙L^−1^ (Figure 22) and increased to approx. 50 g∙L^−1^ in the range of pH = 6.5–11. The significant amount of Na_3_PO_4_ needed to reach pH = 4.5 indicated the precipitation process of the major components of the solution (Figure 21). Figure 23 shows that at pH 4.5, more than 90% of zinc and a part of manganese (about 40%) were precipitated. Increasing the pH above the value of 4.5 did not increase the yield precipitation of zinc, which remained at the level of 93–95%, but it contributed to increase of the yield precipitation of manganese, from 40% at pH = 4.5 to 95% at pH = 6.5. Thus, the use of Na_3_PO_4_ did not lead to selective precipitation of zinc or manganese.

Precipitates were obtained as follows: pH = 4—beige, pH = 4.5–8.5—snow white, pH = 11—light pink. Figure 24 and Figure 25 show the chemical composition and amounts of elements in the precipitates in relation to initial solution volume, respectively, in the function of pH. The precipitate obtained at pH = 4 had a relatively high iron concentration compared to the other precipitates, i.e., 10 wt.%, and the lowest manganese concentration, approx. 4 wt.%. All the precipitates had a similar concentration of phosphorus and zinc, within the ranges of 26–38 wt.% P_2_O_5_ and 47–60 wt.% ZnO, respectively. From the analysis of the composition, it can be concluded that zinc phosphate was the main component of the precipitates. Although precipitation with Na_3_PO_4_ did not contribute to selective zinc-manganese extraction, the results of the research showed that it can be used for iron removal with low manganese losses (4.3% Mn), alkalinizing to pH = 4. Further alkalization, leading to the selective precipitation of zinc, can be carried out with another precipitating agent, e.g., NaOH to a value of 6.5 (as documented previously, for example in Figure 12). It should be mentioned that sodium phosphate is a poorly soluble salt, which may be a disadvantage in its practical application.

After the precipitation by using Na_3_PO_4_, the phosphates phase crystallizes (in the form of eutectic plates) (Figure 26). Contrary to the pictures in Figure 15, the phase separation did not have a (pseudo-eutectic) mixture system, but was a solution, with a high purity zinc phase and had a clearly crystalline form. The precipitation condition with a higher pH was more favorable for the formation of such a structure (i.e., as in Figure 26B,C).

#### 3.3.2. Sodium Carbonate Buffer

Another precipitating agent that has been investigated for use in selective precipitation of zinc from selective treatment (Figure 1) was sodium carbonate (Na_2_CO_3_). Table 8 shows the amounts of Na_2_CO_3_ needed to make the solution alkaline to the adjusted pH (m_Na2CO3_/V_sol_) and includes the amount of solid phase precipitated (∆m_prec_/V_sol_)), and the graphs below are presented on its basis: Figure 27 and Figure 28. Figure 29 shows the precipitation yields of zinc and manganese. During the addition of Na_2_CO_3_ to the solution, gas bubbles escaped from the solution and a characteristic hissing could be noticed: both resulted from the decomposition of carbonate in the acidic environment towards the release of CO_2_. The precipitated solid phases were very fine, which made their filtration from the solution very difficult and almost impossible with the use of even a fine precipitate filter and slow filtration. By analyzing Figure 27, Figure 28 and Figure 29, we can distinguish three stages of alkalization:(1)pH = 6–7—the Na_2_CO_3_ consumption and the resulting amounts of precipitate increased linearly, from about 35 to 53 g∙L^−1^, zinc began to precipitate, and the precipitation yield increased with increasing pH: from 1.22 to 95.3% and some manganese precipitated too, with a minimum yield of 19.5% at pH 6.5;(2)pH 7–9—similarly to the range of pH = 6–7, the Na_2_CO_3_ consumption and the amount of precipitate also increased linearly with increasing pH, moreover, the yields precipitation of zinc remained at a similar level of 94–95%, and the yield precipitation of manganese increased linearly, but not very much, i.e., 74.5–86.7%;(3)pH 9–10—the Na_2_CO_3_ consumption and the amount of precipitate increased significantly, to about 136 g∙L^−1^, the yield precipitation of zinc was still high, and the yield precipitation of manganese reached almost 100%.

**Figure 27 materials-15-03966-f027:**
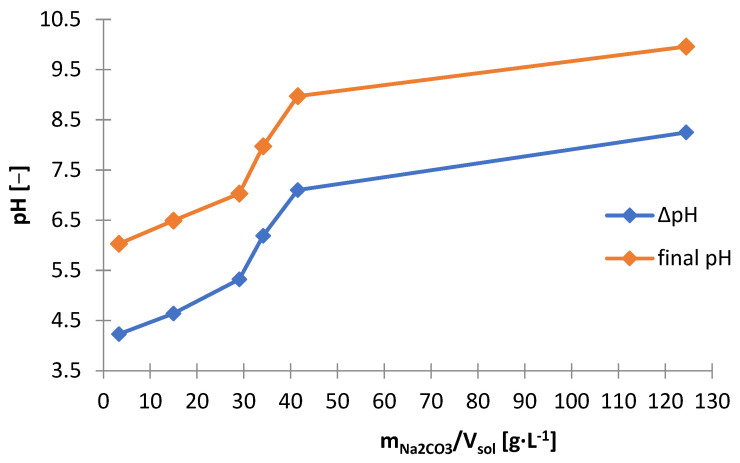
Effect of amount of Na_2_CO_3_ (m_Na2CO3_/V_sol_) on final and change in pH during precipitation from solution after selective leaching.

**Figure 28 materials-15-03966-f028:**
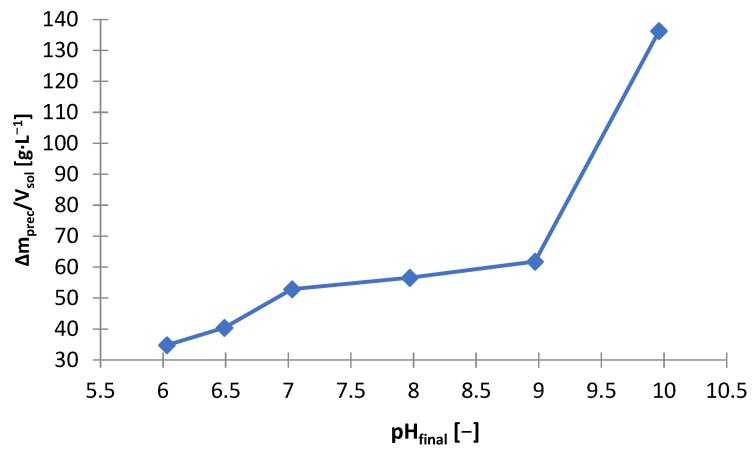
Effect of on final pH (pH_final_) on amount of precipitate (∆m_prec_/V_sol_) during precipitation by using Na_2_CO_3_ from solution after selective leaching.

**Figure 29 materials-15-03966-f029:**
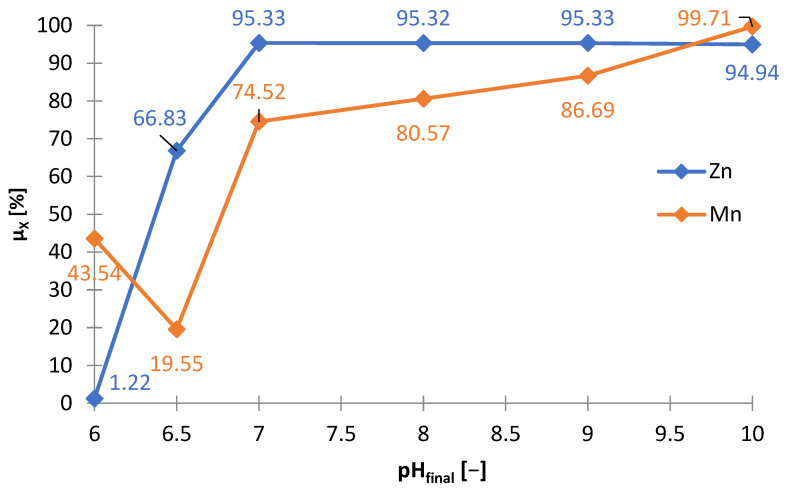
Effect of final pH (pH_final_) on precipitation yields (μ_X_) of zinc and manganese during precipitation by using Na_2_CO_3_ from solution after selective leaching.

**Table 8 materials-15-03966-t008:** Conditions of pH and amount of Na_2_CO_3_ during precipitation of zinc and manganese from solution after selective leaching.

Adjusted pH	Required Amount of Na_2_CO_3_ to Reach Adjusted pH	Temperature of Measurement	Initial pH	Final pH	Change in pH (∆pH)	Amount of Precipitate
-	g∙L^−1^	°C	-			g∙L^−1^
6	3.33	25.48	1.80	6.03	4.23	34.79
6.5	14.96	25.78	1.85	6.49	4.64	40.43
7	29.02	25.32	1.71	7.03	5.32	52.88
8	34.11	25.44	1.78	7.97	6.19	56.58
9	41.50	25.52	1.87	8.97	7.10	61.79
10	124.48	25.53	1.71	9.96	8.25	136.27

The colors of the obtained precipitates were different depending on the final pH: 6—white; 6.5—gray; 7—black; 8–10—from dark to light brown. Figure 30 and Figure 31 show the chemical composition and amounts of elements in the precipitates in relation to initial solution volume, respectively. At pH = 6, the precipitate had the highest concentration of sulfur, i.e., 25.8 wt.%, and a certain content of potassium, manganese and zinc (respectively: 5.4, 11.8 and 55.2 wt.%). The composition changed at pH = 6.5: sulfur—7.2; Mn—4.6; Zn—85.7 wt.%; the precipitate had the highest zinc content at this pH. Increasing the pH (from 6.5 to 10) caused a decrease in the zinc content (from 72.8 to 67.0 wt.%) and an increase in the sulfur content (from 12.0 to 16.6 wt.%).

Figure 32 shows the effects of using sodium carbonate (Na_2_CO_3_). The leakage of gas bubbles from the solution recorded during the tests, caused by the decomposition of this compound into oxidizing agents, translated into the form of very fine precipitates—a compact oxide mixture of pseudo-eutectic nature. The structure was very similar to that obtained directly after using NaOH. The visible lumps of oxide eutectics were, in fact, completely separated compared to the material obtained after NaOH selection (as in the Figure 15), but with a very developed surface—unfavorable for the separation of zinc from manganese, which was recorded in quantitative studies.

Sodium carbonate, just like sodium phosphate, does not allow the selective separation of zinc from manganese, as is the case with the use of sodium hydroxide. Comparing all studied precipitating agents, sodium hydroxide turned out to be the best. Its consumption was the lowest: to reach pH = 6.5 (i.e., to the value at which there is selective zinc separation with minimal co-precipitation of manganese using NaOH exactly, collective separation of zinc and manganese using Na_3_PO_4_ and partial zinc precipitation with the lowest possible manganese co-precipitation, but still significant, when using Na_2_CO_3_) approx. 10 g∙L^−1^ NaOH (Figure 10), approx. 50 g∙L^−1^ Na_3_PO_4_ (Figure 21) and approx. 40.5 g∙L^−1^ Na_2_CO_3_ (Figure 27) were required. Finally, sodium hydroxide should be chosen as the most suitable precipitating agent for the selective precipitation of zinc.

## 4. Summary and Conclusions

The processing of the black mass, obtained from alkaline batteries scrap, by hydrometallurgical method can be summarized as: reductive leaching, alkalization and neutralization, leading to the production of manganese and zinc concentrates. Reductive leaching allows almost all of the manganese and zinc to be leached out of the black mass, i.e., over 99% of zinc and almost 84% of manganese, yielding a graphite concentrate that is a commercially value product. Reductive leaching, however, requires a high liquid–solid ratio, not less than 20 mL∙g^−1^, which may be a problem in implementation of this treatment method on an industrial scale. Leaching tests without the use of a manganese reducing agent show that it is possible to selectively leach zinc without leaching manganese, at an acid concentration of 0.25 mol∙L^−1^; the leaching yield of zinc is 85%, and the leaching yield of manganese is less than 1%, which should be taken into account when investigating the recycling of manganese and zinc from alkaline batteries scrap. With this in mind, the battery recycling process can be carried out in such a way that zinc, as the most valuable component of the battery scrap, is processed as a priority, while the remaining components, i.e., manganese and graphite, as components of less commercial value, are focused on in secondary technological stages.

From the solution containing manganese and zinc, these metals can be separated by alkalization which leads to the precipitation of manganese in the form of manganese hydroxide (Mn(OH)_2_), and then neutralization to precipitate Zn(OH)_2_. Alkalizing requires the consumption of considerable amounts of an alkalizing agent (about 130 g NaOH per liter), which is a significant technological problem. Another problem is the separation of the precipitated manganese hydroxide from the highly alkaline solution (i.e., approx. pH = 13). The precipitated manganese hydroxide takes up a large volume in the solution, which requires the solution to be diluted for easier filtering. The precipitated manganese concentrate is not a pure manganese compound, but it contains significant amounts of zinc (approx. 20 wt.%). Zinc concentrate, from the zinc precipitation step by neutralization using sulfuric acid, is also not pure—it contains approx. 21 wt.% of MnO.

The precipitation of zinc from the solution after the non-reductive leaching of black mass allows the efficient separation of zinc in the form of pure zinc solution, suitable for further processing towards the recovery of zinc, e.g., by electrolysis. Zinc precipitation should take place at pH = 6.5, which requires the use of approximately 10 g NaOH per liter. The precipitation of zinc takes place with over 90% yield while maintaining the minimum precipitation yield of manganese, at the level of several percentage points. This precipitation, compared to the selective precipitation of first manganese and then zinc, requires much less NaOH (like in the case of extraction from collective treatment), and also produces a zinc concentrate of much higher purity (i.e., minimal manganese impurity).

Using sodium phosphate is not an effective method of selective zinc precipitation. Nevertheless, it has been proven that prior to the step of selectively precipitating zinc from manganese, steps may be carried out to purify the zinc-manganese solution with sodium phosphate. The use of sodium phosphate removes the iron with a small loss of zinc and manganese of a few percentage points. This purification can be performed at pH = 4 followed by zinc precipitation with NaOH at pH = 6.5.

Compared to sodium hydroxide, sodium carbonate does not satisfy the same assumptions of the selective precipitation of zinc from manganese from the solution after non-reducing leach. Although the zinc precipitation yields in both cases, and at similar pH values (6.5–7), reach 100%, the use of sodium carbonate at pH = 7 also contributes to the precipitation of manganese in the amount of 74.5%. Moreover, high sulfur concentration is noted in all precipitates obtained as a result of using sodium carbonate (at the level of 7.2–25.8 wt.%). The filtration of these precipitates is difficult because of the very fine size of the solid phase. The above-mentioned disadvantages, i.e., non-selectivity, high sulfur content and problems during filtering, disqualify this reagent in practical application.

These studies have contributed to the increase of practical knowledge on the recycling of zinc and manganese from zinc-bearing batteries scrap by a hydrometallurgical route. The effects of various technological parameters on the leaching/precipitation yields of zinc and manganese were investigated, and the precipitates were quantitatively and qualitatively assessed, the analysis of which is often ignored by some researchers.

## Figures and Tables

**Figure 3 materials-15-03966-f003:**
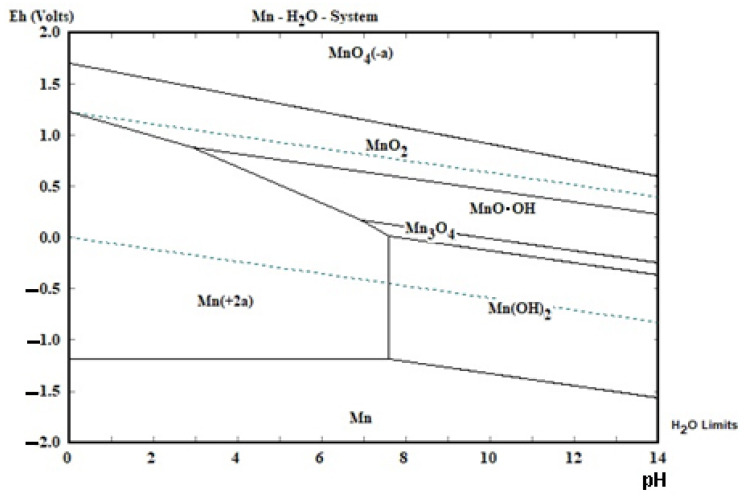
The potential—pH diagram for Mn-H_2_O system at 25 °C and 1 bar for molality 1 mol∙kg^−1^ H_2_O [33].

**Figure 4 materials-15-03966-f004:**
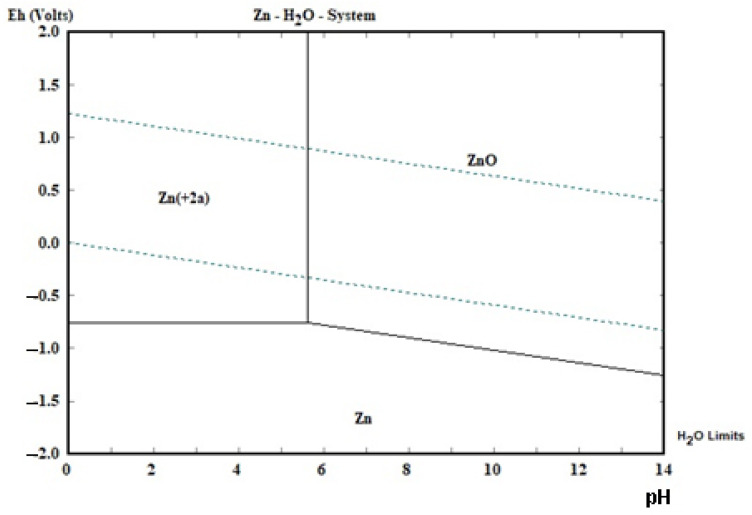
The potential—pH diagram for Zn-H_2_O system at 25 °C and 1 bar for molality 1 mol∙kg^−1^ H_2_O [33].

**Figure 5 materials-15-03966-f005:**
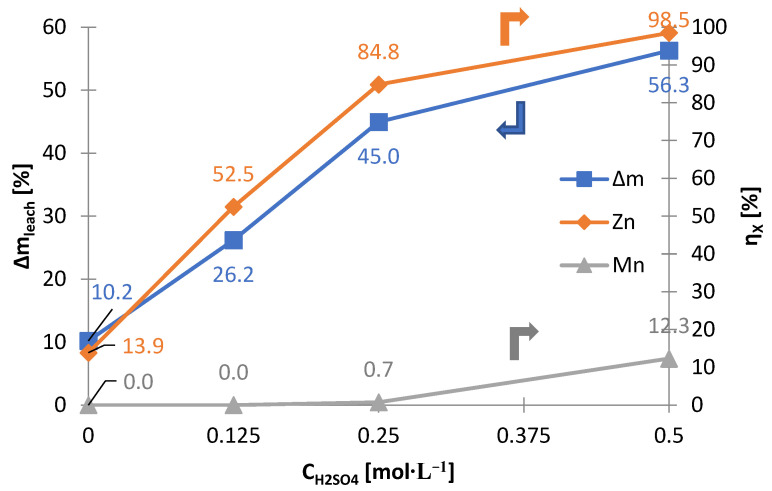
Effects of sulfuric acid concentration (C_H2SO4_) on mass loss of solid phase (∆m_leach_) and leaching yields (η_X_) of zinc and manganese as a result of selective leaching of black mass.

**Figure 6 materials-15-03966-f006:**
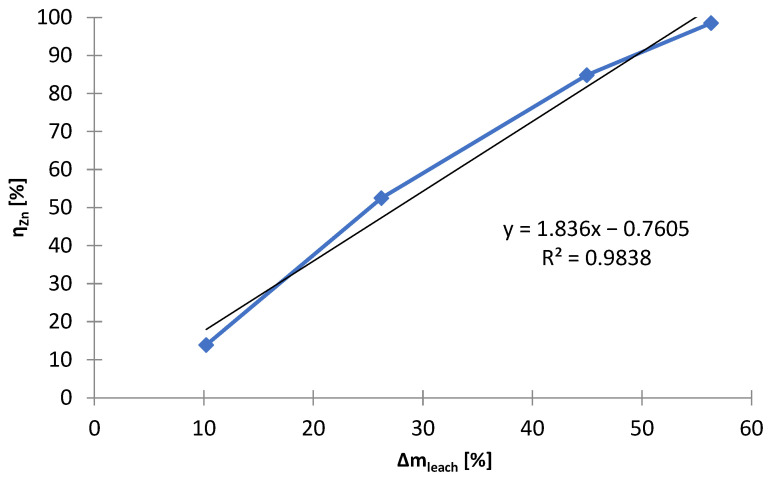
Relationship between ∆m_leach_ and η_Zn_; selective leaching of black mass.

**Figure 7 materials-15-03966-f007:**
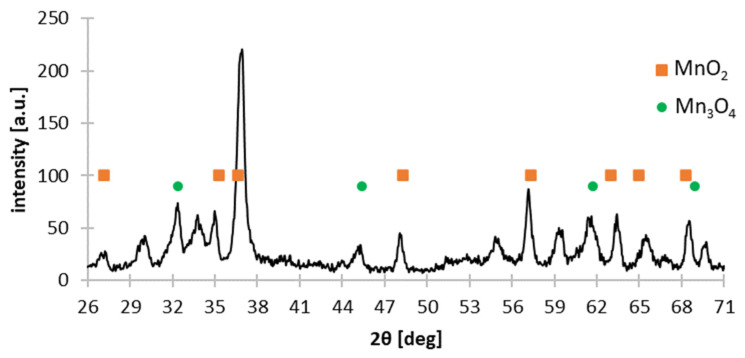
X-ray diffraction pattern of solid residue as a result of selective leaching of black mass in 0.25 mol∙L^−1^ sulfuric acid.

**Figure 8 materials-15-03966-f008:**
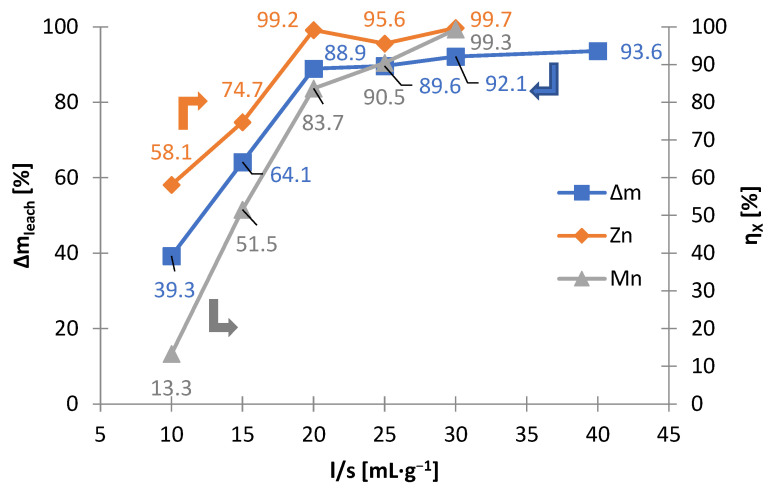
Effects of liquid-to-solid ratio (l/s) on mass loss of solid phase (∆m_leach_) and leaching yields (η_X_) of zinc and manganese as a result of non-selective leaching of black mass.

**Figure 9 materials-15-03966-f009:**
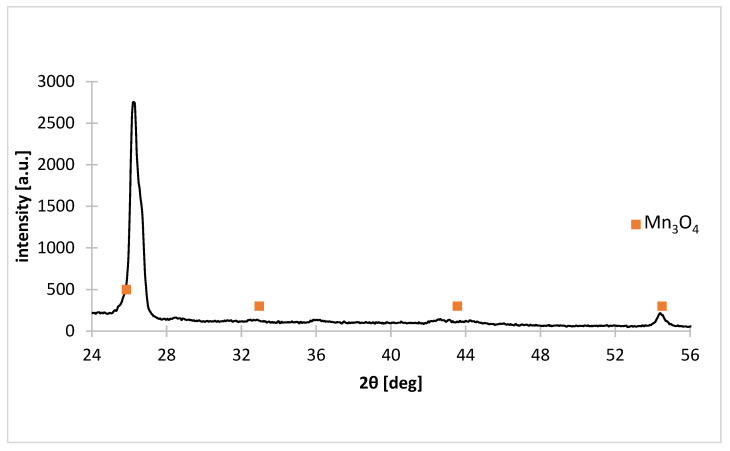
X-ray diffraction pattern of solid residue as a result of non-selective leaching of black mass for liquid–solid ratio = 20 g∙L^−1^.

**Figure 10 materials-15-03966-f010:**
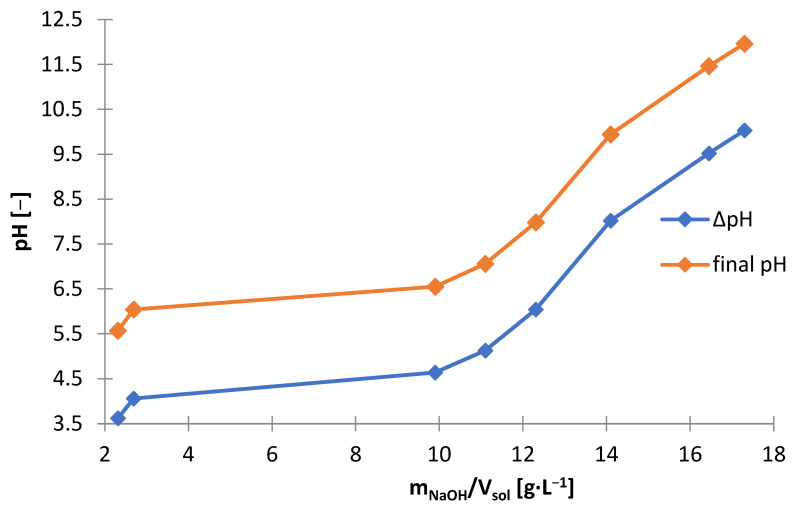
Effect of amount of NaOH (m_NaOH_/V_sol_) on final yield and change in pH during precipitation from solution after selective leaching.

**Figure 11 materials-15-03966-f011:**
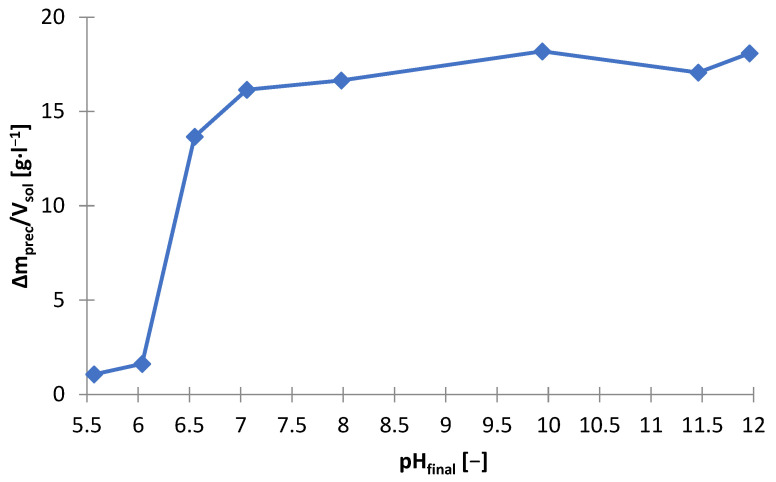
Effect of final pH (pH_final_) on amount of precipitate (∆m_prec_/V_sol_) during precipitation by using NaOH from solution after selective leaching.

**Figure 12 materials-15-03966-f012:**
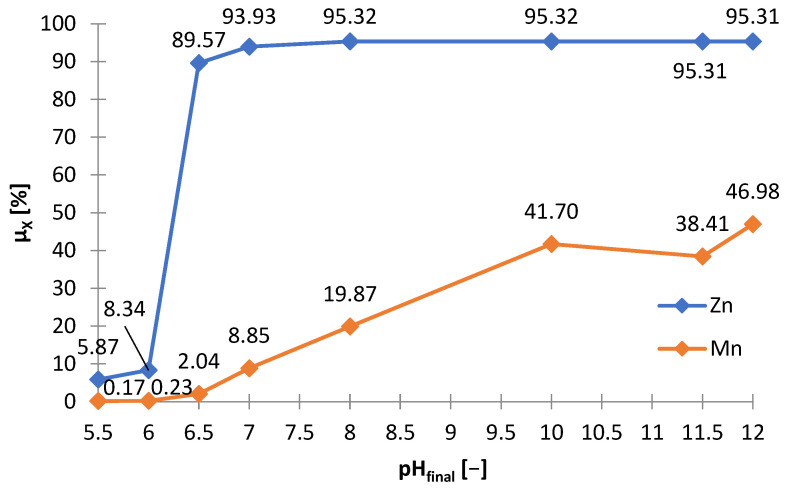
Effect of final pH (pH_final_) on precipitation yield (μ_X_) of zinc and manganese during precipitation by using NaOH from solution after selective leaching.

**Figure 13 materials-15-03966-f013:**
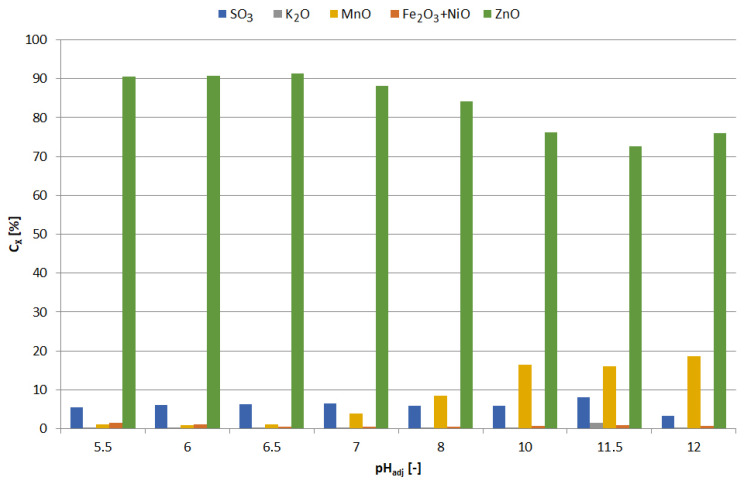
Effect of adjusted pH (pH_adj_) on chemical composition of precipitates as a result of precipitation by using NaOH from selective leaching solution.

**Figure 14 materials-15-03966-f014:**
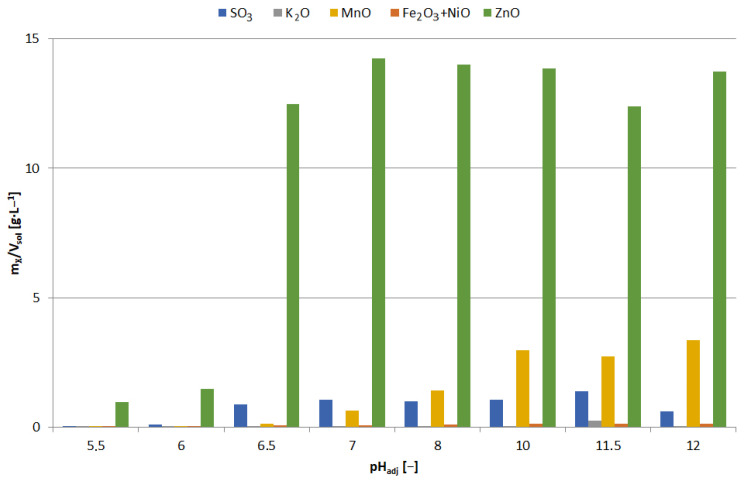
Effect of adjusted pH (pH_adj_) on amount of precipitated compound (m_X_/V_sol_) as a result of precipitation by using NaOH from selective leaching solution.

**Figure 15 materials-15-03966-f015:**
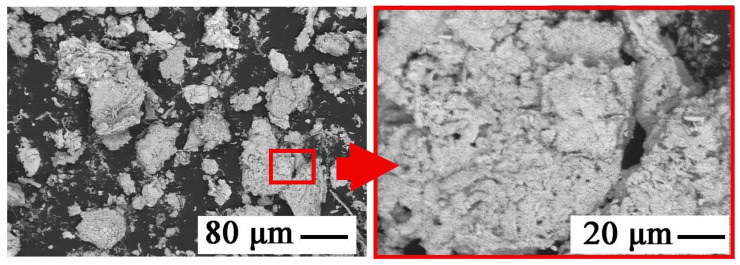
SEM image of the precipitated phase during the selective recycling of black mass from a battery with NaOH as a precipitant at pH 6.5.

**Figure 16 materials-15-03966-f016:**
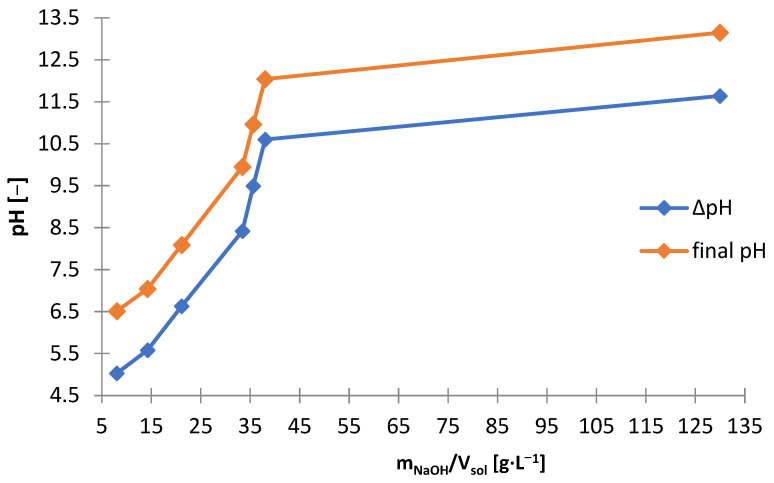
Effect of amount of NaOH (m_NaOH_/V_sol_) on final and change in pH during precipitation from solution after non-selective leaching.

**Figure 17 materials-15-03966-f017:**
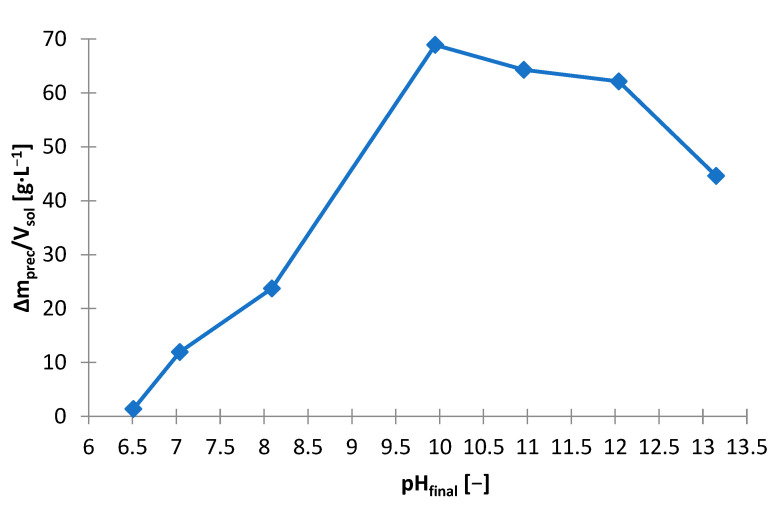
Effect of on final pH (pH_final_) on amount of precipitate (∆m_prec_/V_sol_) during precipitation by using NaOH from solution after non-selective leaching.

**Figure 18 materials-15-03966-f018:**
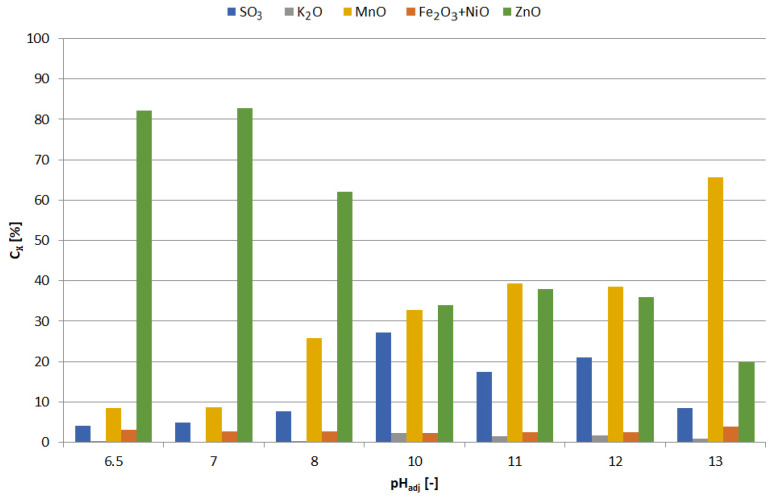
Effect of adjusted pH (pH_adj_) on chemical composition of precipitates as a result of precipitation by using NaOH from non-selective leaching solution.

**Figure 19 materials-15-03966-f019:**
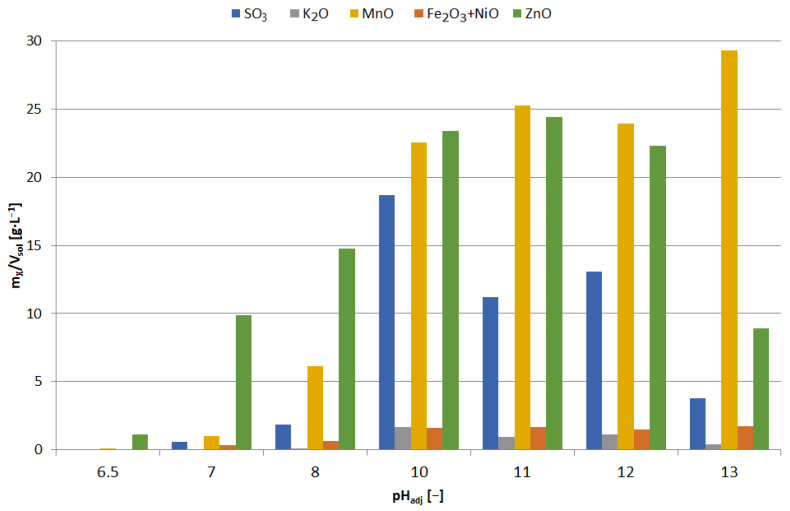
Effect of adjusted pH (pH_adj_) amount of precipitated compound (m_X_/V_sol_) as a result of precipitation by using NaOH from non-selective leaching solution.

**Figure 20 materials-15-03966-f020:**
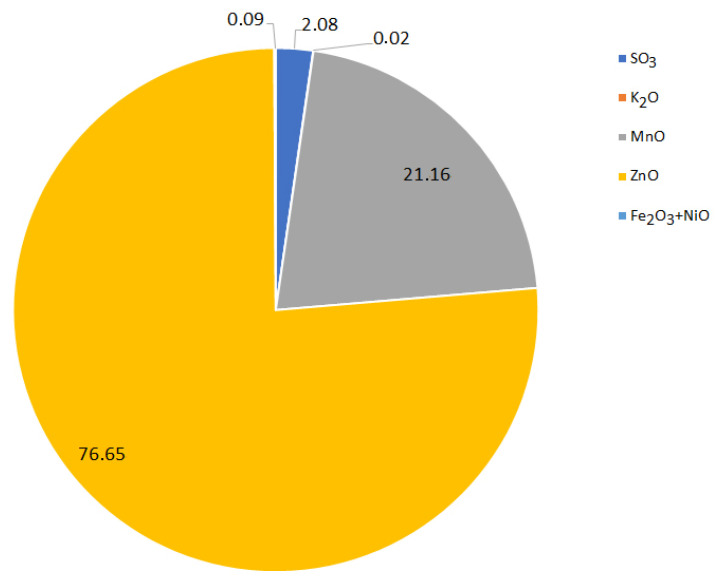
Chemical composition (wt.%) of zinc precipitate as a result of precipitation by using H_2_SO_4_ from solution after manganese precipitation.

**Figure 21 materials-15-03966-f021:**
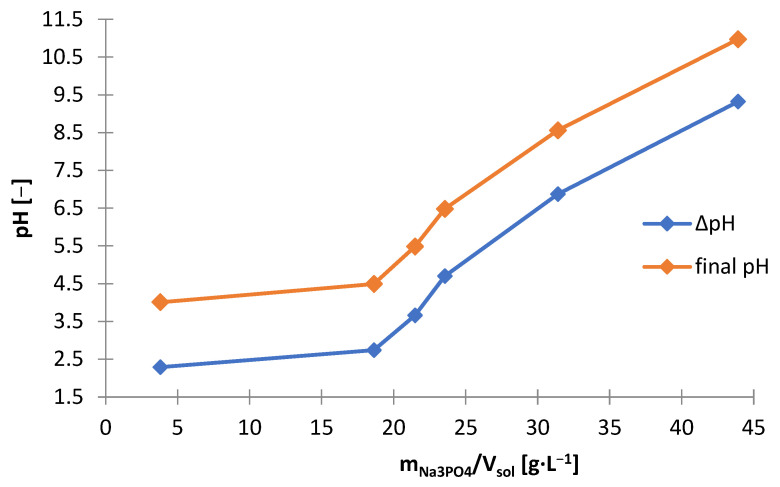
Effect of amount of Na_3_PO_4_ (m_Na3PO4_/V_sol_) on final and change in pH during precipitation from solution after selective leaching.

**Figure 22 materials-15-03966-f022:**
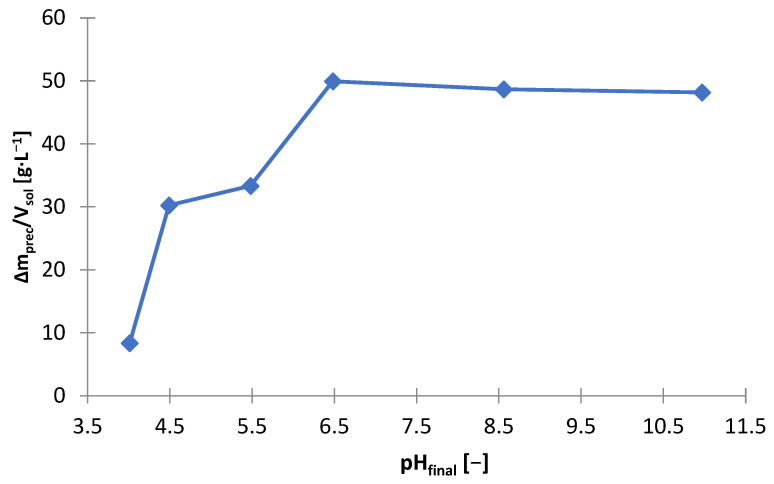
Effect of on final pH (pH_final_) on amount of precipitate (∆m_prec_/V_sol_) during precipitation by using Na_3_PO_4_ from solution after selective leaching.

**Figure 23 materials-15-03966-f023:**
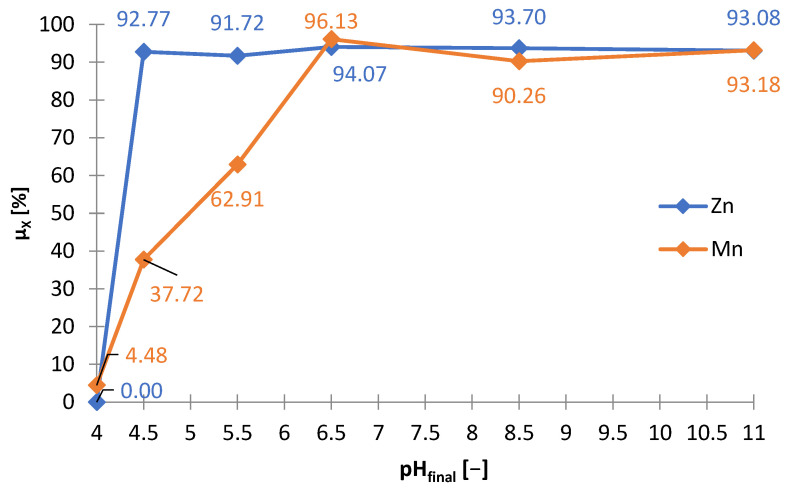
Effect of final pH (pH_final_) on precipitation yields (μ_X_) of zinc and manganese during precipitation by using Na_3_PO_4_ from solution after selective leaching.

**Figure 24 materials-15-03966-f024:**
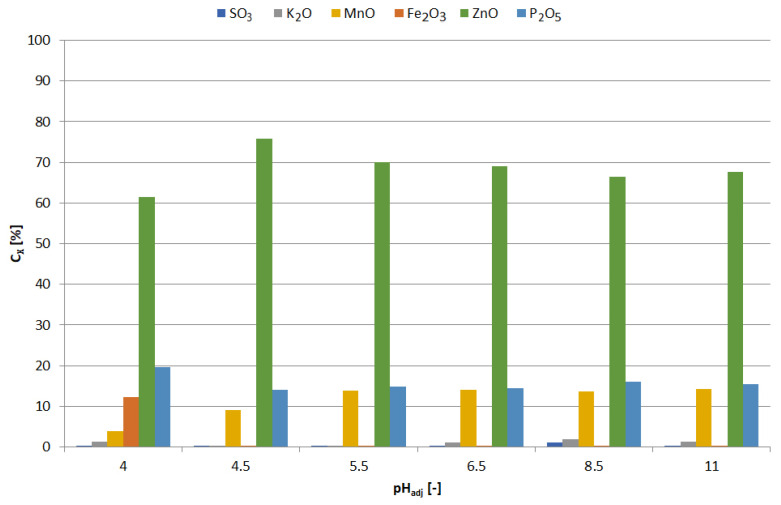
Effect of adjusted pH (pH_adj_) on chemical composition of precipitates as a result of precipitation by using Na_3_PO_4_ from selective leaching solution.

**Figure 25 materials-15-03966-f025:**
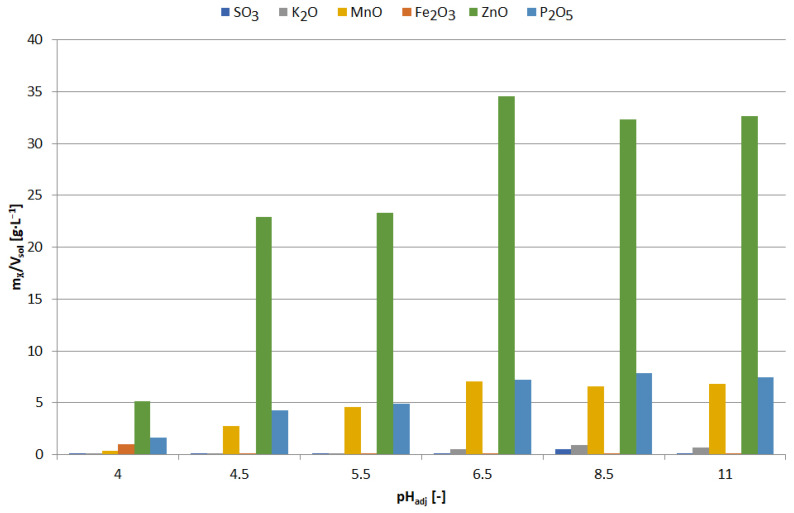
Effect of adjusted pH (pH_adj_) amount of precipitated compound (m_X_/V_sol_) as a result of precipitation by using Na_3_PO_4_ from selective leaching solution.

**Figure 26 materials-15-03966-f026:**
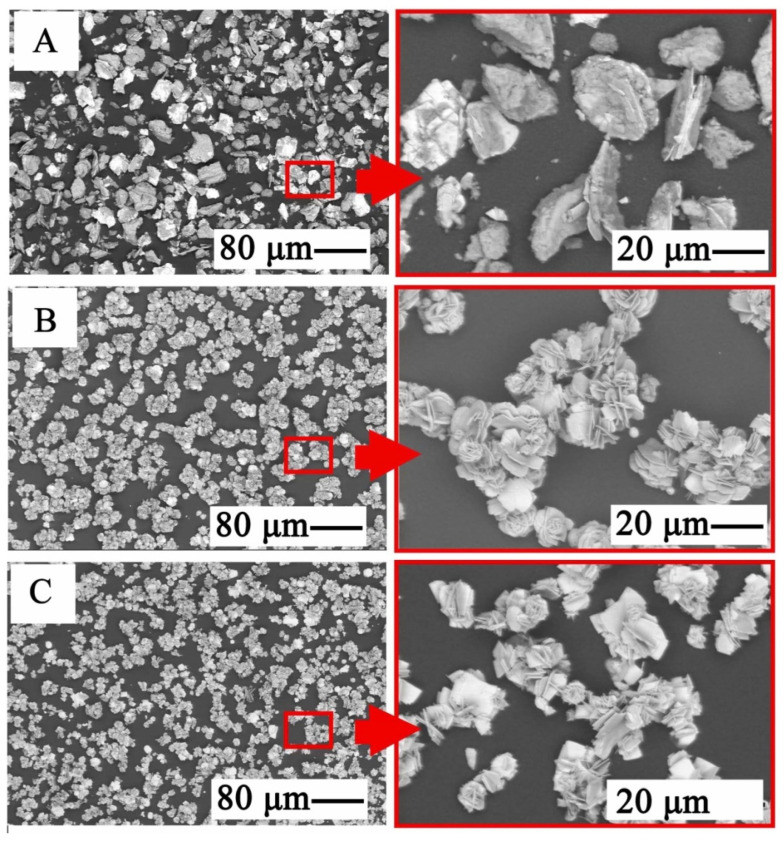
SEM image of the phases precipitated during selective alkalization and zinc precipitation with Na_3_PO_4_ for: (**A**): pH = 4.0, (**B**): pH = 4.5, (**C**): pH = 6.5.

**Figure 30 materials-15-03966-f030:**
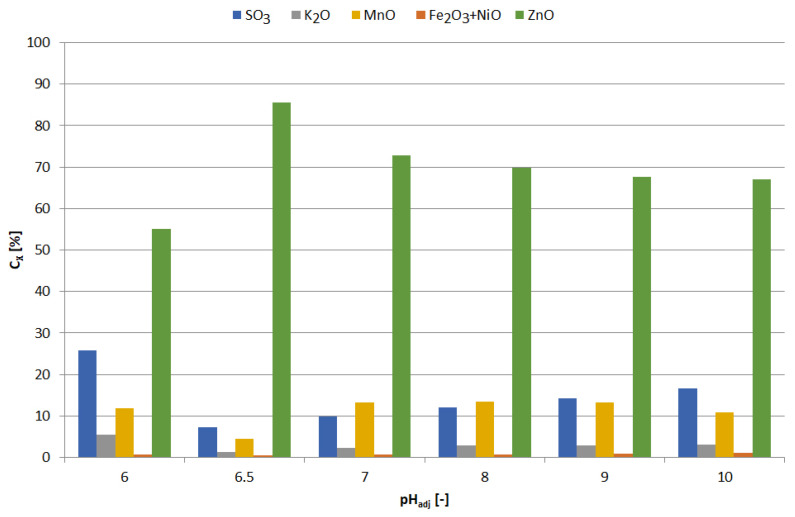
Effect of adjusted pH (pH_adjusted_) on chemical composition of precipitates as a result of precipitation by using Na_2_CO_3_ from selective leaching solution.

**Figure 31 materials-15-03966-f031:**
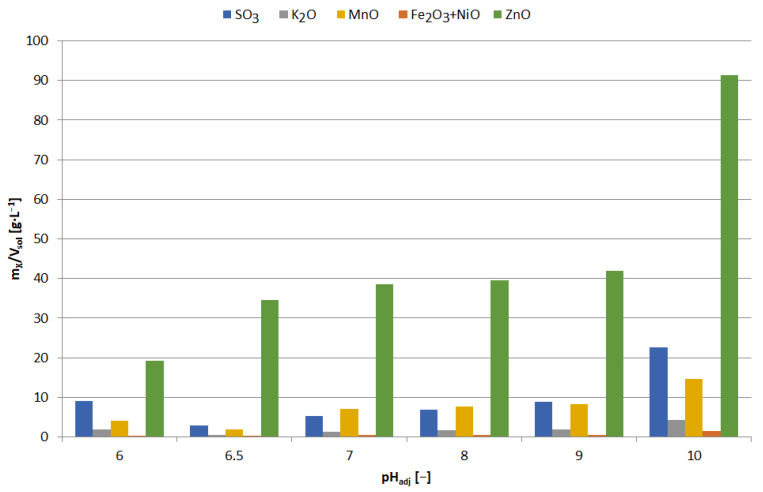
Effect of adjusted pH (pH_adj_) amount of precipitated compound (m_X_/V_sol_) as a result of precipitation by using Na_2_CO_3_ from selective leaching solution.

**Figure 32 materials-15-03966-f032:**
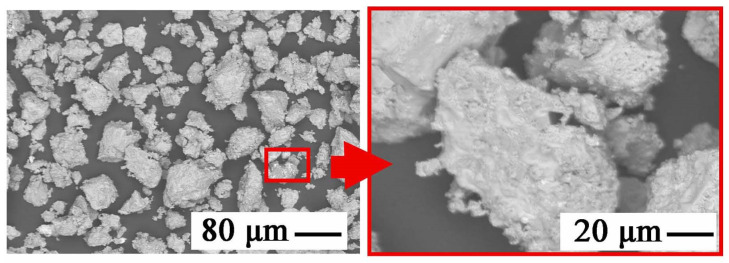
SEM image of the phases precipitated during selective alkalization and zinc precipitation from the electronic mass from the battery with Na_2_CO_3_ participation for pH = 7.0.

**Table 1 materials-15-03966-t001:** Fraction of insoluble solid residue (C_graphite_) in the initial material.

No.	Mass of Initial Material	Mass after Leaching in Aqua Regia	C_graphite_
	g		%
1	10.047	0.626	6.23
2	10.017	0.620	6.19
3	9.9989	0.594	5.94
average		0.613	6.12
standard deviation		0.017	0.16

**Table 2 materials-15-03966-t002:** Chemical composition of the initial material and insoluble solid residue.

		Black Mass Including the Insoluble Solid Residue (Graphite)		Insoluble Solid Residue	
		Average	Standard Deviation	Average	Standard Deviation
P_2_O_5_	% by mass	0.89	0.052	2.23	0.15
SO_3_		0.70	0.014	below the level of detection	
K_2_O		6.49	0.15	0.19	0.01
TiO_2_		0.012	0.0064	1.417	0.01
MnO		42.68	0.36	7.79	4.97
Fe_2_O_3_		0.69	0.070	4.68	1.17
NiO		0.92	0.024	3.75	0.18
CuO		0.033	0.0062	0.35	0.18
ZnO		41.25	0.21	2.14	0.86
BaO		0.21	0.019	0.66	0.11
C_graphite_		6.12	0.16	-	-
SiO_2_		below the level of detection		1.9	0.17
Cl				72.22	4.80
CaO				2.67	0.23

**Table 3 materials-15-03966-t003:** Chemical composition of solid residues as a result of selective leaching of black mass.

		Concentration of Sulfuric Acid			
		0	0.125	0.25	0.5
		mol∙L^−1^			
P_2_O_5_	% by mass	0.77	0.80	0.67	0.69
SO_3_		0.29	0.58	0.62	1.10
K_2_O		1.000	0.417	0.370	0.408
TiO_2_		0.020	0.010	0.036	0.041
MnO		53.32	66.54	81.99	91.22
Fe_2_O_3_		0.925	1.400	2.230	2.920
NiO		0.99	1.15	1.29	1.42
CuO		0.045	0.250	0.100	0.071
ZnO		42.15	28.30	12.10	1.49
BaO		0.24	0.29	0.38	0.42
CaO		0.25	0.21	0.21	0.20

**Table 4 materials-15-03966-t004:** Chemical composition of solid residues as a result of non-selective leaching of black mass.

		Liquid to Solid Phase Ratio (l/s)					
		10	15	20	25	30	40
		g∙L^−1^					
P_2_O_5_	% by mass.	0.84	0.93	0.83	1.30	3.10	2.30
SO_3_		0.54	0.73	1.60	3.90	8.72	9.41
K_2_O		0.280	0.240	0.220	0.280	0.644	0.763
TiO_2_		0.038	0.093	0.160	0.330	0.200	0.530
MnO		64.89	61.45	66.84	51.70	4.02	11.10
Fe_2_O_3_		1.32	3.24	15.90	20.72	32.63	31.74
NiO		1.14	1.26	2.07	2.40	8.52	5.46
CuO		0.062	0.130	4.620	1.380	1.210	2.510
ZnO		30.30	31.00	3.33	1.80	1.55	11.80
BaO		0.39	0.64	4.16	8.67	38.10	23.40
CaO		0.230	0.230	0.300	0.546	1.390	1.090

**Table 5 materials-15-03966-t005:** Conditions of pH and amount of NaOH during precipitation of zinc and manganese from solution after selective leaching.

Adjusted pH	Required Amount of NaOH to Reach Adjusted pH	Temperature of Measurement	Initial pH	Final pH	Change in pH (∆pH)	Amount of Precipitate
-	g∙L^−1^	°C	-			g∙L^−1^
5.5	2.31	21.20	1.95	5.57	3.62	1.07
6	2.69	21.33	1.98	6.04	4.06	1.63
6.5	9.90	21.20	1.91	6.55	4.64	13.66
7	11.10	21.26	1.93	7.06	5.13	16.15
8	12.31	21.08	1.94	7.98	6.04	16.64
10	14.10	21.15	1.92	9.94	8.02	18.19
11.5	16.45	21.32	1.94	11.46	9.52	17.06
12	17.30	21.25	1.93	11.96	10.03	18.08

**Table 6 materials-15-03966-t006:** Conditions of pH and amount of NaOH during precipitation of zinc and manganese from solution after non-selective leaching.

Adjusted pH	Required Amount of NaOH to Reach Adjusted pH	Temperature of Measurement	Initial pH	Final pH	Change in pH (∆pH)	Amount of Precipitate
-	g∙L^−1^	°C	-			g∙L^−1^
6.5	8.02	28.46	1.48	6.51	5.03	1.37
7	14.23	29.10	1.46	7.04	5.58	11.93
8	21.14	29.08	1.46	8.09	6.63	23.75
10	33.42	28.57	1.53	9.95	8.42	68.91
11	35.62	29.06	1.47	10.96	9.49	64.29
12	38.02	28.25	1.44	12.04	10.60	62.13
13	129.97	28.35	1.51	13.15	11.64	44.62

**Table 7 materials-15-03966-t007:** Conditions of pH and amount of Na_3_PO_4_ during precipitation of zinc and manganese from solution after selective leaching.

Adjusted pH	Required Amount of Na_3_PO_4_ to Reach Adjusted pH	Temperature of Measurement	Initial pH	Final pH	Change in pH (∆pH)	Amount of Precipitate
-	g∙L^−1^	°C	-			g∙L^−1^
4	3.80	22.47	1.72	4.01	2.29	8.35
4.5	18.63	22.29	1.75	4.49	2.74	30.23
5.5	21.48	22.84	1.82	5.48	3.66	33.32
6.5	23.55	22.67	1.78	6.48	4.70	49.94
8.5	31.40	23.04	1.69	8.56	6.87	48.65
11	43.90	22.84	1.65	10.97	9.32	48.17

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
