# Peer review of "Studies of Selective Recovery of Zinc and Manganese from Alkaline Batteries Scrap by Leaching and Precipitation"

_materials, 2022, doi:10.3390/ma15113966_

Round 1

Reviewer 1 Report

Dear authors

The present manuscript called "Studies of selective recovery of zinc and manganese from alkaline batteries scrap by leaching and precipitation" seems to me to be a good study. However, the manuscript requires improvements prior to publication.

I think I should mention the general context of the problem at the beginning of the introduction, the search for raw materials in alternative sources or recycling, and why it is so beneficial for the planet. (DOI: 10.1016/j.mineng.2022.107441) these problems are mentioned here.

Why did you consider ascorbic acid for the reduction mechanism and not another reducing agent?

Perhaps you could say something like "There are various reducing agents with good results for the acid-reductive dissolution of manganese such as  SO2 (DOI: 10.1016/0883-2927(94) 00050-G), hydroxylammonium chloride (DOI: 10.1002/jctb.3948), etc. However, ascorbic acid was selected due to....."

Correct the ml unit throughout the document, the correct one is mL

I think the equation presented in item 2.1 is too obvious. However, I leave it to your discretion whether to keep this or not.

You should improve the presentation of Table 2.

The results and discussions are good and well written. However, these results need to be contrasted with other similar studies (very few comparisons are mentioned in the manuscript).

In his conclusions, he compares a conventional leaching where practically no manganese is recovered (although a lot of Zinc), compared to a reducing one with good results in both elements.
I think that this paragraph fails to mention the low commercial value of manganese, compared to other elements present in the battery, which could only make it attractive if the process is cheap and very efficient at the same time.
Finally, I think you should choose an alternative or give a proposal, this can be at the end of the conclusions.

Regards

Author Response

Dear Reviewer 1:

  1. Information of the search for raw materials in alternative sources or recycling, and why it is so beneficial for the planet has been added to the article (highlighted yellow)
  2. Why did you consider ascorbic acid for the reduction mechanism and not another reducing agent?->Ascorbic acid has the advantage that it is easy to make a solution (it is a solid, free from molecular water, easy to use and easy to calculate). In citations 27 and 29, they used ascorbic acid as the best of the reducing agents - and so we did
  3. Improve of table 2 -> I don t see what should be improved
  4.  ...results need to be contrasted with other similar studies->Our article is not a review, we present our results. If we were to confront the results of other authors now, the article would be about 50 pages long, which is probably not the way it works. Moreover, some precipitants were used by me for the first time, so  it can t be compare to others

Reviewer 2 Report

Dear authors, looking at the article you are offering, are not too clear about the purpose of the studies. It remains the impression that the proposed methodology does not solve the task that you have set up, namely  recycling and extracting Zn and Mn from alkali batteries. A large number of studies have been carried out and definite and interesting results have been obtained. Therefore, at least in the conclusions, you should also highlight the practical detail of the studies, which I think is the most important in your work.

Author Response

Dear Reviewer 2

1. The practical detail of the studies has been added in the conclusions (  in the attachment)

Reviewer 3 Report

It is a very interesting paper.

In the abstract , line 13, write: is the liquid/solid treatment(l/s) ratio...

page 4, eq:4: not 102, but 10 MnSO4 sol.

page 7, line 207: not Cx,2, but Cx,0 (author: see as in eq: 11).

page 14, lines 336 and 338: from selective or collective treatment.

page 14, line 344: after selective (fig.1) leaching is presented...

page 15, fig. 11: Effect of final pH, not “of on”. 

page 17, line 336: fig. 15b, not fig. 16b.

page 17, line 393: collective leaching (fig. 2). 

page 20, lines 429-434, 434-441, 444-448: too long sentences.

page 20: line 451: from collective treatment (fig. 2).

page 20, line 450: is not very effective (fig. 2), which...

page 21, line 468: selective leaching (fig. 1). The possibility...

page 21: line 466, create a sub-chapter: “3-3-1- Sodium phosphate buffer”.

page 24, lines 520-522, rebuild the sentence: Fig. 26, the area... 

page 25, line 527: create a sub-chapter “3-3-2 Sodium carbonate buffer”.

page 25 line 528: selectively treatment (fig. 1) was sodium carbonate. 

page 27, lines 561-562, write: pH 6: white, pH 6.5: gray, pH 7: black, pH 8-10: dark to light brown.

page 28, line 576, delete: "another zinc precipitating agent -".

page 28, lines 589-595: sentence too long.

Author Response

Dear Reviewer 3:

Your suggestions have been added to the article (in the attachment)

Reviewer 4 Report

Comments

The selective recovery of zinc and manganese from alkaline batteries scrap by leaching and precipitation was systematically studied in this manuscript. Recovery of zinc and manganese from the alkaline batteries scrap was carried out in the continuous steps. The products of the selective or collective leaching of black mass highly depend on the pH value of the solution at different steps. The critical technological parameter of l/s ratio and the optimal H2SO4 concentration was given, which should be at least 20 ml∙g‐1 and 0.25mol∙l‐1, respectively. In the collective leaching process, 95.6‐99.7% of Zn and 83.7‐99.3% of Mn were leached. The manuscript was well organized and the experiment results were abundant, which offers an alternative method to recover effectively the zinc, manganese, and graphite precipitation from the black mass. The manuscript can be accepted after minor revision and the detailed comments are as follows:

  1. As a general note, I recommend double-checking the whole manuscript by a native speaker, because some parts of the manuscript are hard to understand or are written with vocabulary/grammar mistakes.
  2. The “l/s ratio” in the abstract should be substituted by the full tile “liquid‐solid ratio” because it was firstly mentioned in the manuscript.
  3. The scale bar of the X and Y axis should be added in figures 5, 6, 7, 8, etc. as figures 3 and 4. The figures should be clear to express the scientific means independently.
  4. In the equations, the symbols of the elements, such as Na, O, and S, should be standard form, not italic.

Author Response

Dear Reviewer 4:

  1. The entire manuscript has been re-checked by a native speaker
  2. The l/s ratio was  substituted by the full tile liquid‐solid ratio
  3. The scale bar of the X and Y axis has been added
  4. The symbols of the elements, such as Na, O, and S is in standard form

See  the attachment

Round 2

Reviewer 1 Report

Dear authors

Hello, the suggested corrections were made. I recommend this manuscript for publication

Regards

Author Response

  I made the suggested corrections to the article ( in the attachment)
